# Exploring the Core Microbiota of Four Different Traditional Fermented Beverages from the Colombian Andes

Johannes Delgado-Ospina [1,2,*], Luisa Fernanda Puerta-Polanco [1], Carlos David Grande-Tovar [3],
Raúl Alberto Cuervo [1], Diana Paola Navia-Porras [1], Luis Gabriel Poveda-Perdomo [1],
Fabián Felipe Fernández-Daza [4] and Clemencia Chaves-López [2,*]

[1] Grupo de Investigación Biotecnología, Facultad de Ingeniería, Universidad de San Buenaventura Cali, Carrera 122 # 6-65, Cali 76001, Colombia

[2] Faculty of Bioscience and Technology for Food, Agriculture and Environment, University of Teramo, Via R. Balzarini 1, 64100 Teramo, Italy

[3] Grupo de Investigación de Fotoquímica y Fotobiología, Universidad del Atlántico, Carrera 30 # 8-49, Puerto Colombia 081008, Colombia

[4] Grupo Microbiología, Industria y Ambiente (GIMIA), Universidad Santiago de Cali, Calle 5 # 62-00, Cali 760035, Colombia

* Correspondence: jdelgado1@usbcali.edu.co (J.D.-O.); cchaveslopez@unite.it (C.C.-L.)

**Abstract:** Fermentation is an ancient process used to prepare and preserve food. Currently, fermented beverages are part of the culture of people living in the Colombian Andean Region, and they are a vital part of their cosmology and ancestral vision. Chicha, Forcha, Champús, and Masato are some of the most common Colombian Andes region's traditional fermented beverages. These drinks come from the fermentation of maize (*Zea maize*), but other cereals such as wheat or rye, could be used. The fermentation is carried out by a set of bacteria and yeasts that provide characteristic organoleptic properties of each beverage. In this work, the information collected from the metagenomics analyses by sequencing ITS 1-4 (Internal Transcriber Spacer) and the 16S ribosomal gene for fungi and the V3-V4 region of the rDNA for bacteria allowed us to identify the diversity present in these autochthonous fermented beverages made with maize. The sequencing analysis showed the presence of 39 bacterial and 20 fungal genera. In addition, we determined that only nine genera of bacteria and two genera of fungi affect the organoleptic properties of smell, colour, and flavour, given the production of compounds such as lactic acid, alcohol, and phenols, highlighting the critical role of these microorganisms. Our findings provide new insights into the core microbiota of these beverages, represented by *Lactobacillus fermentum*, *Acetobacter pasteurianus*, and *Saccharomyces cerevisiae*.

**Keywords:** fermented beverages; microbiota; maize; yeast; lactic acid bacteria; acetic acid bacteria

## 1. Introduction

Fermentation is one of the first and most economical methods for food preservation [1]. It is a slow transformation bioprocess of organic compounds induced by microorganisms or enzymes that converts carbohydrates to alcohol, organic acids, and other molecules [2]. In addition to preservation, fermentation adds benefits such as new flavors, digestibility, and pharmacological and nutritional values [3–6].

In the manufacture of fermented beverages, raw materials such as fruits and cereals are the sources of sugars and fermentable starches. As a result of fermentation, typically driven by lactic acid bacteria and yeasts, multiple bioactive metabolites such as antioxidant compounds, vitamins, peptides, essential amino acids, and fatty acids are generated, allowing an increase in the nutritional properties of these fermented products [7].

Indigenous fermented foods and beverages have been prepared for centuries at home or in the cottage industry using simple techniques and equipment. Today, they remain essential for the well-being of many people [8]. The fermentation process is spontaneous and

uncontrolled. The products are often obtained under local climatic conditions, significantly heterogeneous depending on the traditions and cultural preferences, availability of raw materials, and geographical areas of production [9].

Fermented foods and beverages are an integral part of the cultural heritage of Colombia. This country has a rich diversity of fermented beverages; however, most food practices are regional and confined to a specific community. Information on the microbial community and physicochemical parameters during the spontaneous fermentation of traditional fermented beverages in the Andes region of Colombia is scarce [8]. Around the world, the interest in cereal-based fermented products is increasing due to their low fat/cholesterol, high minerals, phytochemicals, and dietary fibre content [3]. Archaeological data suggests that maize drinking existed in Mesoamerica long before the Classic Period [10]; also, in the Andes, these types of fermented beverages had a dual function of food and ceremony [11], and have been reported to exist as far back as the Inca period. In Colombia, maize grains are the central ingredient in several fermented beverages, including Chicha, Forcha, Champús, and Masato. These products' colour, texture, and taste depend primarily on adding sugar cane derivates, fruits, microbial species and fermentation time (Figure 1).

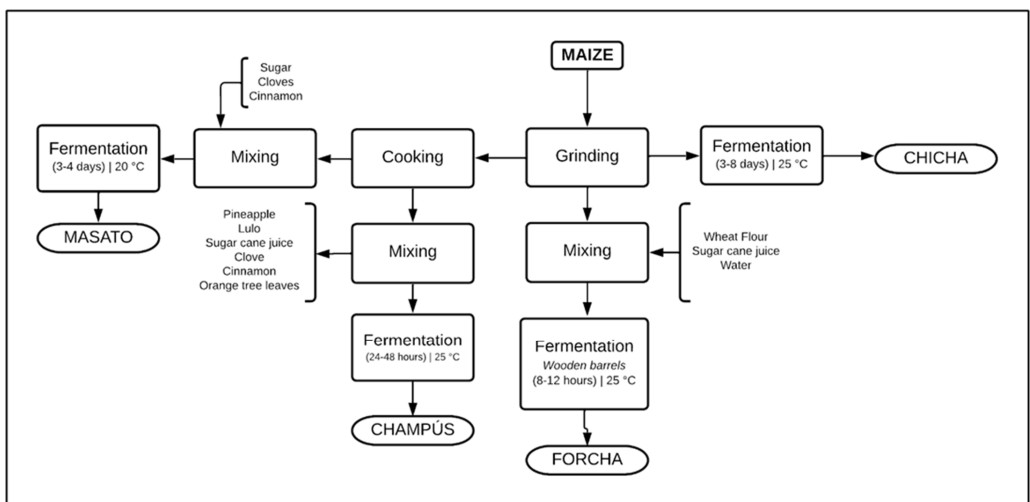

**Figure 1.** Flowchart of the production of four traditional fermented beverages from the Colombian Andes (Chicha, Forcha, Champús, and Masato) elaborated from maize.

In this regard, Chicha is a light brown drink of liquid consistency made from the fermentation of cooked maize (3 to 8 days) [8]. Forcha is a white drink with a foamy consistency produced from the natural fermentation of uncooked maize or wheat flour, sugar cane and water in wooden barrels for 8 or 12 h [12]. Masato is made from the fermentation of cooked maize with the addition of sugar cane, cloves and cinnamon, resulting in a beverage with light brown colour and thick consistency, with a time of fermentation of between 3 and 4 days [13]. Champús is a fermented beverage made of cooked maize and other ingredients such as pineapple, lulo (*Solanum quitoense* Lam.), sugar cane syrup, clove, cinnamon, and orange tree leaves, and the fermentation is carried out over 24 and 48 h [8,13,14]. The microbiota of some of these fermented products has been studied by culture-dependent methods [8,14,15]. However, these methods underestimate the composition and diversity of the microorganisms [16]. Herein for the first time, metagenomic sequencing was used to investigate the bacterial and fungal community of the aforementioned fermented beverages. It is well known that culture-independent techniques enhance the monitoring of microbial diversity, structure, and dynamics in naturally fermented foods. In addition to the dominant microorganisms, these advanced technologies can reveal the rarer bacterial and fungal genera to elucidate their possible role in fermentation. The results of our study are essential in promoting the appreciation and safeguarding of these indigenous beverages as part of Colombia's intangible cultural heritage.

## 2. Materials and Methods

### 2.1. Samples

In this study, the traditional fermented beverage samples were obtained from local producers in the Andes region in Colombia. Samples of Chicha were collected in Cauca Department, while those of Forcha in Quindío Department, those of Champús in Valle del Cauca Department, and those of Masato in Cundinamarca-Tolima Department. In this work we take in consideration the best-known local producers which have been producing the drinks by natural fermentation for many years. Three samples were collected for each beverage and refrigerated at 4 °C until analysis. Under refrigerated conditions, these beverages remain drinkable for one week, except for Chicha, which remains acceptable for about two weeks.

### 2.2. Physicochemical Characterization

The soluble solids measurement was made with an Abbe refractometer (Atago DR-A1, Tokyo, Japan) at 25 °C. The pH was determined directly (Orion 2 Star, Thermo Scientific, Waltham, MA, USA). For all determinations, samples consisted of 25 mL of beverage homogenized in a vortex for 1 min and filtrated with a cotton canvas filter.

The total solids content was determined from 5.0 g of sample, dried at 65 °C for 24 h. After that time, the resulting sample was incinerated at 600 °C for three hours in a furnace for ash determination.

The Kjeldahl method was performed according to method 992.23 of the AOAC International in a heat block (DK6, Velp Scientifica, Monza, Milano, Italy) at 420 °C for 2 h. The conversion factor was 6.25 to determine total protein content [17]. Three replicates of each of the three samples of each beverage were performed.

### 2.3. Antioxidant Capacity Assays

Sample preparation was carried out according to Chaves et al. [18] with some modifications. For the first extraction, 25 mL of samples were mixed with 5 mL of sulfuric acid (0.005 M) and centrifuged at 10,000 rpm/10 min/25 °C, and the supernatant was recovered. The residue was extracted with acetone/water (70:30 $v/v$) and centrifuged again under the conditions previously described. The supernatants were diluted in water to determine antioxidant activity (ABTS and DPPH and total phenol index (TPI)). Three replicates of each of the three samples of each beverage were performed.

#### 2.3.1. ABTS Radical Cation (ABTS•+) Scavenging Activity Assay

The radical scavenging activity was measured by the ABTS radical cation discolouration assay [19]. In brief, supernatants were successively diluted in deionized water. Next, 30 μL of the diluted sample was mixed with 2.97 mL of ABTS radical solution and left in the dark for 6 min. Absorbance was determined at 734 nm using a spectrophotometer (Genesys 10 UV, Thermo electron, Waltham, MA, USA). Radical scavenging activity was expressed as μmol of Trolox equivalent per litre beverage.

#### 2.3.2. DPPH Radical Scavenging Ability Assay

The radical scavenging ability of the supernatants was measured using the stable radical DPPH. The procedure described by Villa-Rodríguez et al. [20] was used with some modifications. In brief, the DPPH solution was adjusted at $0.70 \pm 0.02$ of absorbance at 515 nm. For the reaction: in a test tube, 2.9 mL of DPPH radical and 100 μL of the extract properly diluted (successively diluted) was added. Absorbance determination occurred after 30 min in the dark. The results were compared with the calibration curve of Trolox and expressed as μmol of Trolox equivalent per litre of beverage.

#### 2.3.3. Total Phenol Index

The Total Phenol Index (TPI) used the Folin-Ciocalteu method [21] with some modifications. In brief, a 100 μL of diluted samples (or standard for the calibration curve) were

added: 250 μL of Folin-Ciocalteu reagent 1N (dissolved in distilled water from the commercial reagent 2 N), 1250 μL of sodium carbonate 20% (*w/v*), and 1400 μL of water. Once prepared, the mixture remained in the dark for two hours. Absorbance was determined at 760 nm using a spectrophotometer. The results were compared with the calibration curve of gallic acid and expressed as mg of gallic acid equivalent per litre of beverage.

### 2.4. Organic Acid, Ethanol, and Sugar Content

The samples were homogenized with equal volume (25 mL) of 5 mM $H_2SO_4$ in a disperser (T25 digital, IKA, Staufen, Germany) at 15,000 rpm for 30 s. The samples were then centrifuged at $10,000 \times g$ for 10 min at 4 °C. Five extractions were obtained from each beverage's samples.

Saccharose, glucose, fructose, lactic acid, and ethanol were analyzed with HPLC equipment coupled with a refractive index detector L-2400 (Elite Lachrom, Hitachi, Tokyo, Japan). Twenty microliters of sample were injected into a column (Agilent Hi-Plex H, 300 mm × 6.5 mm, 8 μm) using 5 mM $H_2SO_4$ as mobile phase. Peaks were identified and quantified by comparison with adequate standards.

### 2.5. Microbiological Analyses

Ten grams of beverage were homogenized in a Stomacher Lab-blender in a 90 mL sterile saline solution. Decimal dilutions of the suspension were plated and incubated as follows: Lactic Acid Bacteria (LAB) in MRS agar (Oxoid, Basingstoke, UK) supplemented with 250 mg/L of fluconazole (Sigma-Aldrich Italy, Milan, Italy) at 37 °C in anaerobiosis for 48 h. Acetic Acid Bacteria (AAB) in GYC agar (20 g/L glucose, 10 g/L yeast extract, 3 g/L $CaCO_3$, 15 g/L agar, 70 g/L ethanol) supplemented with 250 mg/L of fluconazole at 37 °C for 72 h, and those that showed halos of degradation were considered positive. Yeasts in YPD agar (Biolife, Italiana, Milan, Italy) were added with 150 ppm chloramphenicol (Sigma-Aldrich Italy, Milan, IT) at 25 °C for 24 h, and moulds in DG18 Agar (Oxoid) and Czapec- Agar (Biolife) were supplemented with 150 ppm chloramphenicol for 96 h. Total coliforms were isolated in Violet Red Bile Glucose Agar and Violet Red Bile Agar (Oxoid, Basingstoke, UK) at 37°C for 24 h, respectively, in anaerobiosis. The visible colony count at the end of the incubation period and the dilution factor were used to determine the number of microorganisms in the sample.

Three replicates of each of the three samples of each beverage were performed.

### 2.6. Genomic DNA Extraction

Genomic DNA was isolated using a QIAamp Powerfacal DNA kit (Qiagen, Carlsbad, CA, USA). Briefly, 100 mL of each beverage was homogenized in a paddle blender (Stomacher 400, Seward Ltd., Worthing, UK) for 10 min. The liquid was centrifuged at $170 \times g$ at 4 °C for 5 min. For DNA extraction and purification, the supernatant (0.25 mL) was taken and continued according to the manufacturer's instructions.

DNA Sample Quality and Quantity Specification

The OD260/280 ratio was used as an indicator of sample purity; the samples that reached values of 1.8–2.0 and a minimum concentration of 100 ng/μL were sequenced. Additionally, gel electrophoresis was performed to assess the status of the DNA sample.

The DNA was sent to Macrogen (Macrogen, Seoul, Republic of Korea) for sequencing. The V3 and V4 regions of 16S rDNA and ITS region gene products from each beverage were sequenced on an Illumina HiSeq platform (Metagenome Amplicon Sequencing: MiSeq/HiSeq2500). The Illumina sequencer generates raw images utilizing sequencing control software for system control and base calling through integrated primary analysis known as RTA (Real Time Analysis). The BCL (base calls) binary is converted into FASTQ utilizing the illumina package bcl2fastq.

Scythe (v0.994) and Sickle programs were used to remove the adapter sequences. After adapter trimming, reads shorter than 36 bp were dropped to produce clean data.

### 2.7. Sequence Analysis and Species Identification

A bioinformatic analysis was performed with QIIME 2 version 2020.8 (https://view.qiime2.org/ Flagstaff, AZ, USA) (accessed on 1 November 2022) [22], using the q2-demux algorithms (for paired sequences), followed by DADA2 (https://benjjneb.github.io/dada2/ CC-BY 4.0)(accessed on 01 November 2022) [23] for artefact removal and selection of quality sequences (Quality Score ≥ 20). The taxonomic assignment used the untrained Bayesian classifier [24] and Greengenes 13_8 99% OTU reference sequences for 16S [25] and UNITE for ITS [26]. The reads that could not be assigned to the specific genus level were allocated as unclassified taxa.

### 2.8. Alpha Diversity Analysis

Alpha diversity indices were calculated using the QIIME at the OTU level. The overall richness was calculated using the Chao1 richness estimator, and the overall diversity was calculated using the Shannon index [27]. Venn diagrams were realized based on the OTUs obtained for the different species.

### 2.9. Statistical Analysis

Statistical analysis of the physicochemical characterization and microbial population data was conducted using the Kruskal–Wallis tests and the multiple nonparametric comparisons test at a level of $p < 0.05$ using the Statistica 12.0 program (2013) (StatSoft, Dell, Austin, TX, USA). The relative abundance of each OTU was determined for each sample, and the differences between the calculation of the sample were analyzed with a Student's t-test. All values are shown as means with the standard deviation.

Hierarchical clustering analyses were performed using the ClustVis web server. Data were standardized in percent relationship versus the number of OTUS in each beverage and transformed into ln(x + 1) before clustering. Beverages were clustered using correlation distance and average linkage [28].

## 3. Results

Microbial counts (Table 1) revealed that the viable microbiota was dominated by the association of presumptive LAB and yeasts, together with acetic bacteria, particularly in the Chicha and Forcha beverages. Coliforms were undetectable (<1.0 log CFU/g) in the analyzed samples. The low counts of yeast and LAB in Chicha could be due to the long fermentation time of the beverage which increased the ethanol content to 9.9%. These results are consistent with Elizaquível et al. [29].

From the microbial counts it seems that Forcha and Masato fermentations were guided by LAB that reached levels of $6.10 \pm 0.4$ Log CFU mL$^{-1}$ and $5.2 \pm 0.2$ Log CFU mL$^{-1}$ respectively, while Champús beverages were guided by yeast which were detected at levels of $6.8 \pm 0.2$ Log CFU mL$^{-1}$.

**Table 1.** Microbial counts and pH measured in the different Colombian fermented beverage samples.

| | Fermented Beverage | | | |
|---|---|---|---|---|
| | **Chicha** | **Forcha** | **Champús** | **Masato** |
| Lactic acid bacteria (Log CFU mL$^{-1}$) | $2.4 \pm 0.3$ a | $6.10 \pm 0.4$ c | $3.8 \pm 0.1$ ab | $5.2 \pm 0.2$ bc |
| Acetic bacteria (Log CFU mL$^{-1}$) | $2.3 \pm 0.2$ a | $2.1 \pm 0.1$ a | <1.0 | <1.0 |
| Yeast (Log CFU mL$^{-1}$) | $5.1 \pm 0.3$ b | $4.2 \pm 0.4$ ab | $6.8 \pm 0.2$ b | $2.1 \pm 0.2$ a |
| Coliforms | <1.0 | <1.0 | <1.0 | <1.0 |

Results are expressed as means ± standard deviations of three independent samples, each with three replicates. Different letters in the same row indicate significant differences ($p < 0.05$).

### 3.1. Sequence Analysis

The proportions of high-quality sequences were above 79.7% in all beverages. The number of high-quality sequences ranged between 260,053 and 338,532 for bacteria and

297,640 and 405,662 for fungi, indicating that the sequences obtained were sufficient to represent the microbial structure of the beverage samples. Bacteria comprised 79.76%, 86.23%, 85.40%, and 86.14% of assigned reads across the different beverages (Masato, Forcha, Champús, and Chicha), respectively, and fungi made up 84.29%, 82.94%, 87.87%, and 85.46% of the assigned reads, respectively.

### 3.2. Bacterial Communities of the Different Fermented Beverages

The analyses allowed us to detect 11 bacterial phyla in the different fermented beverages, including Firmicutes, Proteobacteria, Cyanobacteria, Verrucomicrobia, Armatimonadetes, Bacterioidetes, Chloroflexi, Thermotogae, Chrysiogenetes, Actinobacteria, and Tenericutes. The phylum Firmicutes and Proteobacteria were common in all the beverages. In particular, Firmicutes were the predominant phyla, with 98% relative abundance in Forcha, Chicha, and Champús, and 70% in Masato (Figure 2). Proteobacteria was the subdominant phylum with a relative abundance of 30% in Masato compared to 1.5% in Chicha and 1.19% in Champús. The remaining phyla were present only in Forcha beverage, accounting for only 1.12% of the bacterial community. Table S1 exhibits taxonomic abundances in Forcha of <1%.

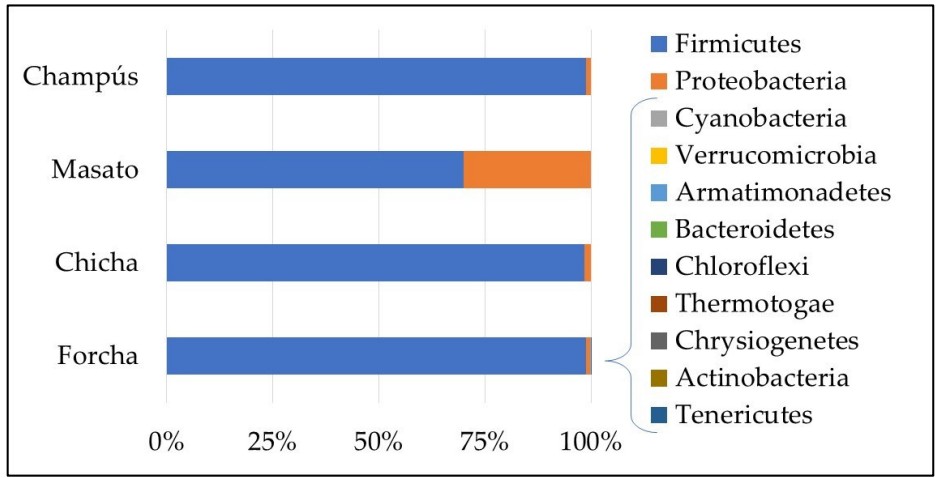

**Figure 2.** Relative abundance (%) of bacterial 16S rDNA genes from the Colombian maize fermented beverages at the phylum level.

The bacterial reads of the Forcha were further clustered into 11 classes, 17 families, 30 genera, and 56 species. Meanwhile, those of Chicha corresponded to two classes, four families, seven genera, and 16 species. As regards Champús following the sequence classification, the bacteria reads were clustered in six classes, eight families, 12 genera, and 12 species. On the other hand, the bacterial reads of Masato were clustered in three classes, three families, three genera, and nine species.

The percentages of bacterial OTUs assigned to the genus level are depicted in Figure 3. The most abundant and frequent bacterial genera detected belonged to the Firmicutes phylum. In fact, in the Forcha, Chicha, and Masato beverages, *Lactobacillus* was predominant, although their presence was variable among these beverages (81.9%, 89.5%, and 70.0%, respectively). While in Champús beverage, the *Weissella* genus was the predominant one, accounting for 68.5%. In addition, the subdominant genus varied according to the type of beverage. For example, *Oenococcus* was subdominant (16.93%) in Forcha while *Acetobacter* (29.6%) was subdominant in Masato and *Lactobacillus* (29.4%) was subdominant in Champús. Other genera with minor abundances are indicated in Table S2.

The species-level taxonomic analysis showed that *Lactobacillus sanfranciscensis*, *Oenococcus oeni*, *L. reuteri*, *L. buchneri*, and *L. fermentum* were predominant in Forcha samples. However, in Masato samples, the proportion of the most abundant species comprised *L. fermentum*, *L. ruminis*, *L. reuteri*, and *Acetobacter pasteurianus*. On the other hand, Champús

samples were characterized by the abundance of *L. fermentum, Weisella koreensis, L. buchneri, L. brevis, L. reuteri, and A. pasteurianus* while in Chicha samples, only two species *L. lactis* and *W. koreensis* were found to be the most abundant (Figure 4). Table S3 shows the minor species with a relative abundance of less than 1.0%.

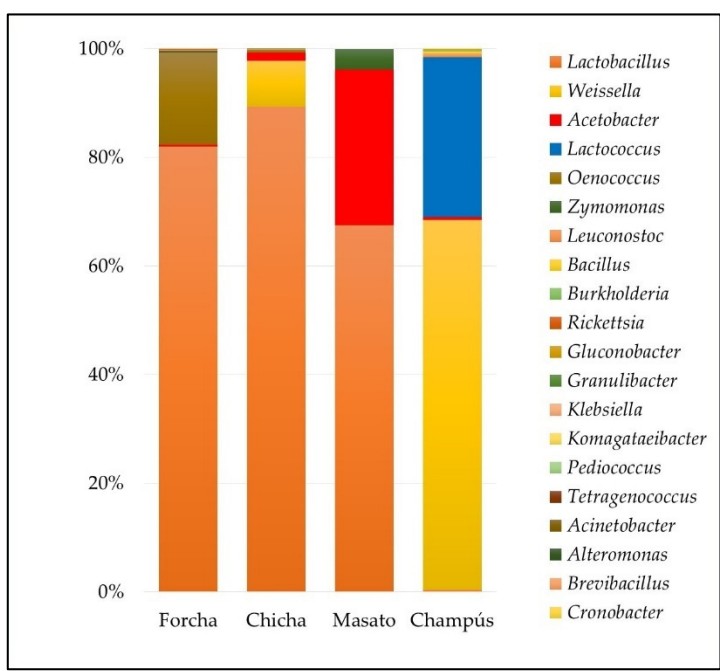

**Figure 3.** Relative abundance (%) of bacterial 16S rDNA genes from the Colombian maize fermented beverages at the genus level (top 20 genus).

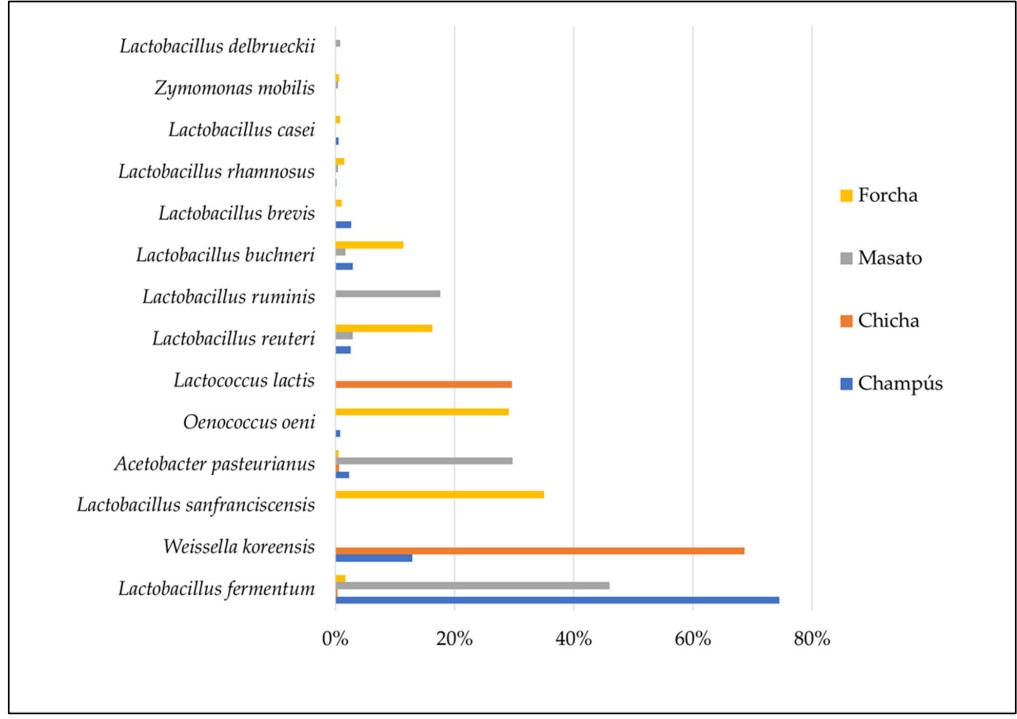

**Figure 4.** Relative abundance (>1%) of species-level taxon from each beverage.

### 3.3. Fungal Communities of the Different Fermented Beverages

A fungal community composition analysis of the data set classified only Ascomycota and Basidiomycota fungal phyla in the various fermented beverages. In particular, Basid-

iomycota was detected in Masato only, with 2.13% relative abundance. At the family level, Saccharomycetaceae was the dominant family in all the samples.

The distribution of the fungal genus is shown in Figure 5. The dominant genera are associated with alcoholic fermentations, generally producing alcohol through the fermentation of different carbohydrates. *Saccharomyces* were predominant in Forcha and Chicha (77.6% and 84.8%, respectively); instead, *Pichia* was the predominant one in Masato (24.8%) and *Hanseniaspora* was predominant in the Champús sample (66.46%). The genera *Deekera* (19.82%), *Pichia* (23.17%), *Candida* (16.77%), *Clavispora* (9.45%), *Meyerozyma* (9.45%), and *Geotrichum* (7.01%), were the subdominant populations in the Masato beverage, Pichia (9.2%) and *Starmera* (9.9%) were the dominant ones in Champús. Other subdominant genera are listed in Table S4.

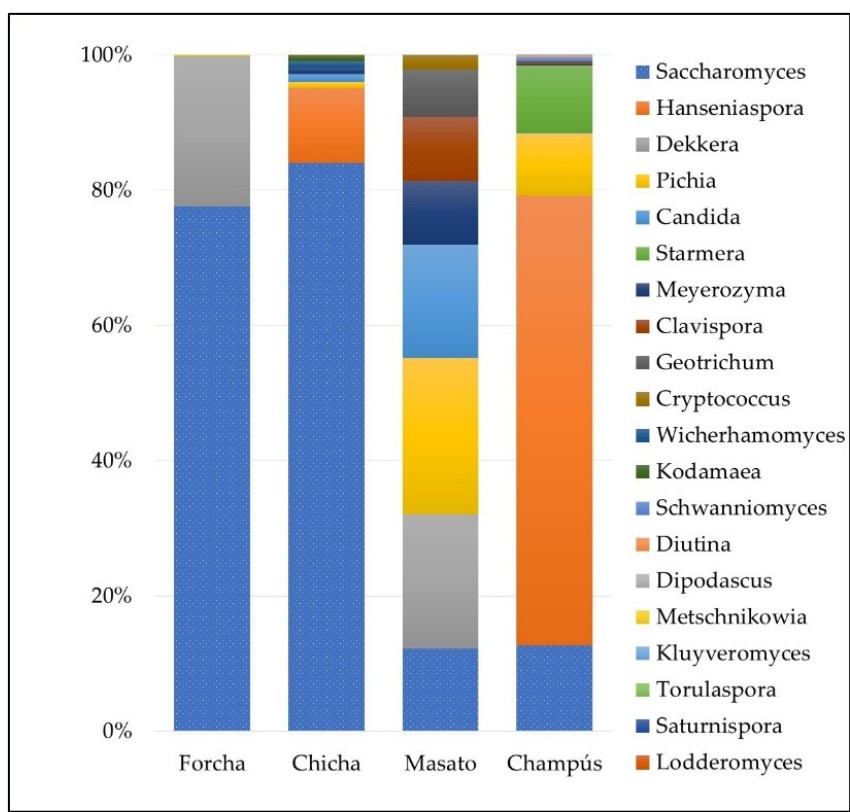

**Figure 5.** Relative abundance (%) of fungal ITS region genes from the Colombian maize fermented beverages at the genus level (Top 20 genus).

As regard the specie level, *Saccharomyces cerevisiae* was the most common yeast, accounting for 77.6%, 84.79%, 12.2%, and 12.68% of the Forcha, Chicha, Masato, and Champús beverages, respectively (Figure 6). *Dekkera bruxellensis* and *Pichia kudriavzevii* were the next most common species, in particularly *D. bruxellensis* accounted for 21.24% in Masato and 22.31% in Forcha of the relative abundance, while *P. kudriavzevii* accounted for 5.0% of Champús and 24.84% of Masato. *Meyerozyma guillelmondi* and *Clavispora Lusitania* were found in Masato, Chicha, and Champús, and were also identified from the fungal sequence reads but in low percentages. *Hanseniaspora nectarophila* (66.48%) and *Starmera stellimalicola* (9.92%) were present only in Champús. In addition, *Candida tropicalis* (7.19%), *Geotrichum candidum* (7.52%), and *C. famata* (10.78%) were found only in Masato and *H. uvarum* was found only in Chicha. We also identified fungal species not yet associated with maize fermentations, such as the plant pathogenic filamentous fungus *Fusarium oxysporum* which was detected in Champús; the human pathogenic *C. tropicalis* and the Basidiomycetes *Cryptococcus humicola* frequently reported as soil species [30] detected in Masato samples.

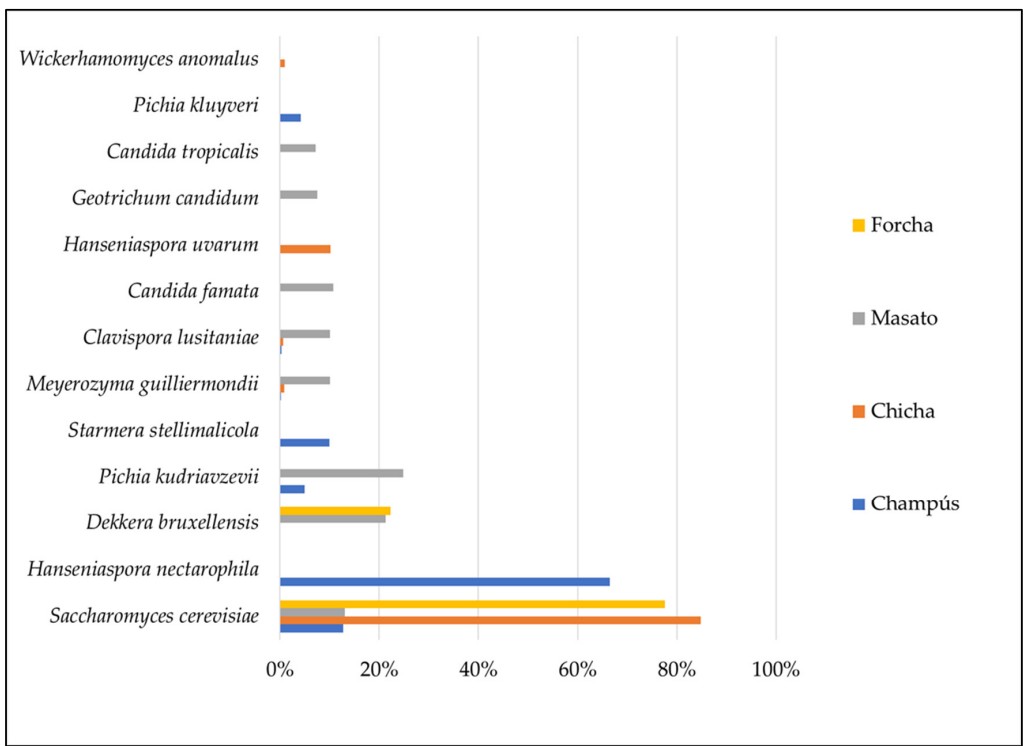

**Figure 6.** Relative abundance of fungal communities in the four fermented beverages at the species level, determined by metagenomic analysis of the ITS region.

*3.4. Species Richness and Diversity Index*

Regarding taxonomic classification at the species level, 61 bacterial species and 34 fungal species were identified. Species richness (observed species and Chao1) and diversity (Shannon and Simpson) indices were used to characterize the α-diversity of the microbial community in the different fermented beverages, separating them into bacteria and fungi. In general, the analysis revealed a more significant bacterial diversity than in fungi (Table 2).

**Table 2.** Alpha diversity indexes of the microbial community in fermented beverages.

| Fermented Beverages | Observed Species | | Chao1 | | Shannon | |
|---|---|---|---|---|---|---|
| | Bacteria | Fungi | Bacteria | Fungi | Bacteria | Fungi |
| Chicha | 16 | 11 | 21 | 13 | 1.42 | 0.89 |
| Forcha | 56 | 4 | 72 | 4 | 2.41 | 0.77 |
| Champús | 12 | 20 | 14 | 21 | 1.04 | 1.66 |
| Masato | 9 | 9 | 14 | 11 | 1.88 | 2.85 |

Regarding bacteria, the central microbial richness was observed in the Forcha beverage, which was consistent with the results of the average number of reads and OTUs, probably because during the manufacturing of this drink, uncooked maize flour is used and because fermentation is carried out in wooden barrels containing an inoculum from the precedent fermentation. While in Chicha, Champús, and Masato, the lowest richness values were found to be attributable to the initial cooking of some of its ingredients in the production process and to the fact that fermentation is currently carried out with metallic utensils that have replaced traditional clay utensils where they were previously made. Regarding fungi, a greater richness was observed in Champús because, during the process of making this beverage, a greater quantity of ingredients such as fruits (pineapple, lulo) and orange leaves are added after the cooking process. The fruits contribute to the fermentative process primarily via yeasts [14].

The biodiversity index of Shannon revealed that bacteria diversity was in the following order: Forcha > Masato > Chicha > Champús, while that of fungi was Masato > Champús > Chicha > Forcha, which suggests that the differences in row materials, fermentation process and environment mainly affect the microbial richness and microbial diversity.

### 3.5. Similarities and Differences among the Microbial Communities of the Fermented Beverages

Venn diagrams were plotted to demonstrate similarities among microbial communities in the different beverage samples in terms of overlapping OTUs (Figure 7A). Only two OTUs were shared by all the maize fermented beverages samples. In particular, *L. fermentum* and *A. pasteurianus* were present in all of the samples. Moreover, Forcha and Champús beverages showed 36 and 4 unique OTUs, respectively, as evidenced by the non-overlapping part of the Venn diagram. In contrast, Masato and Champús did not show any specie that characterizes these beverages.

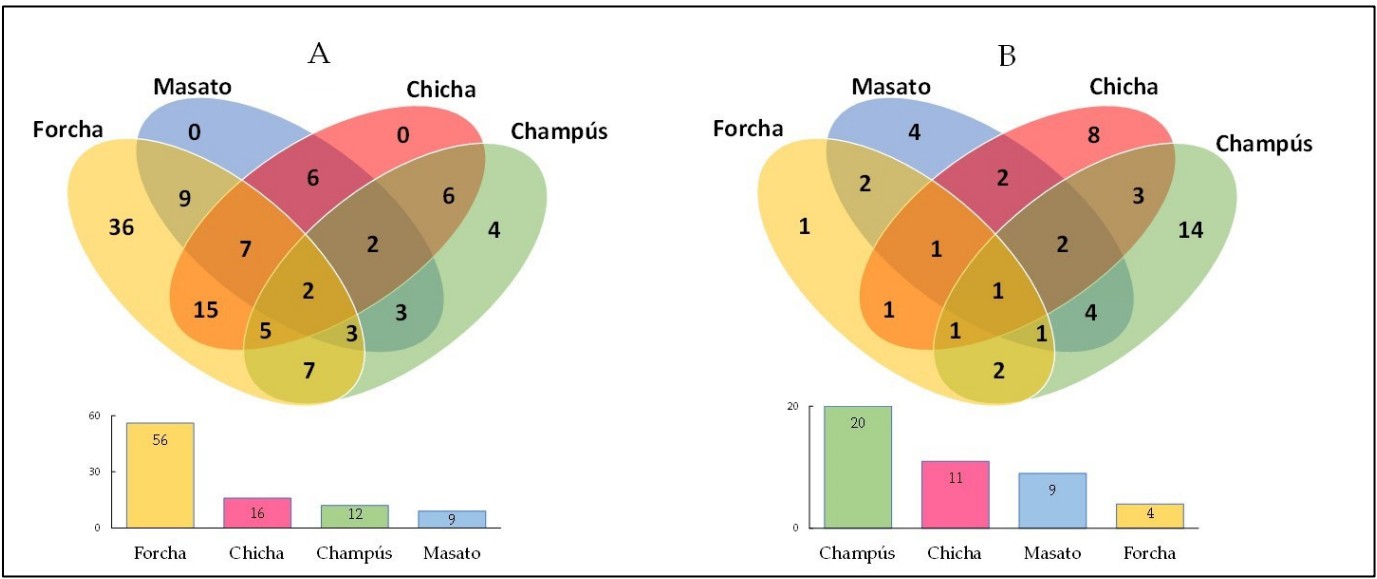

**Figure 7.** Venn diagram showing species similarity between the fermented beverage samples in terms of OTUs overlapping. (**A**) bacterial and (**B**) fungal.

Regarding fungi, the Venn diagram showed that the four beverages only share one specie at the OTUs level (base call accuracy 99.9%), which indicated that these beverages had a low level of similarity in fungal species. However, these beverages were characterized by a significant number of species compared with bacteria; in addition, Champús, Chicha, and Masato possessed 14 out 20, 8 out 11, and 4 out 9 species characterizing each beverage (Figure 7B and Table S5).

### 3.6. Dominant Microbiota Identification

A hierarchical clustering analysis was used to determine if the microbial structure was similar between the different beverages. For bacteria, genera grouping comparisons divided the samples into two groups; the first group consists of Forcha, Chicha, and Masato, mainly due to the contribution of the genus *Lactobacillus* (Figure 8). In the case of fungi, three groups were found, one group containing Chicha and Forcha, mainly due to the contribution of the *Saccharomyces* genus, thus it was found abundantly. The second group was Masato, characterized by the genera *Candida, Geotrichum,* and *Pichia*. The third group corresponded to Champús; in this group, a more significant number of dominant genera were found but were mostly unrelated to the other beverages. In general, the microbiota found in Champús is different from that found in the other beverages.

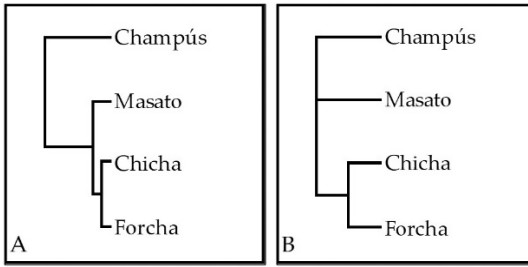

**Figure 8.** Hierarchical clustering analysis based on genus level of the bacterial (**A**) and fungal (**B**) communities.

*3.7. Chemical Characteristics of the Beverages*

Table 3 shows the mean values and standard errors of the characteristics measured in the four different fermented beverages studied here. We identified significant differences among beverages. As observed, the fermented beverages presented a low pH consistent with the content of lactic acid, with Chicha the most acidic (3.50 ± 0.08) and Masato the less acidic (4.35 ± 0.15). Chicha also showed a high alcohol content (9.9%), while in Masato and Champús, the alcohol production was deficient.

**Table 3.** Characterization of fermented beverages.

| Parameters | Fermented Beverage | | | |
| --- | --- | --- | --- | --- |
| | Chicha | Forcha | Champús | Masato |
| pH | 3.50 ± 0.08 a | 3.76 ± 0.75 ab | 3.92 ± 0.15 ab | 4.35 ± 0.15 b |
| Lactic acid (g/L) | 16.7 ± 1.7 c | 8.8 ± 3.0 bc | 5.5 ± 2.8 ab | 0.3 ± 0.1 a |
| Ash (%) | 2.17 ± 0.82 b | 0.46 ± 0.21 a | 0.96 ± 0.25 a | 0.53 ± 0.01 a |
| Protein (%) | 5.30 ± 0.7 b | 1.41 ± 0.3 a | 1.12 ± 0.1 a | 3.40 ± 0.6 ab |
| Saccharose (g/L) | 4.8 ± 2.9 a | 28.1 ± 10.3 b | 127.2 ± 18.8 c | 117.1 ± 1.8 c |
| Glucose (g/L) | 2.9 ± 2.4 a | 23.0 ± 11.5 b | 24.0 ± 16.4 b | 3.2 ± 0.1 a |
| Fructose (g/L) | 14.1 ± 7.2 b | 2.8 ± 1.1 a | 21.1 ± 14.0 b | 1.0 ± 0.1 a |
| Soluble solids (%) | 8.5 ± 1.5 a | 26.3 ± 7.6 c | 16.0 ± 4.0 bc | 12.5 ± 2.5 ab |
| Ethanol (%) | 9.9 ± 1.2 c | 2.9 ± 1.1 bc | 0.5 ± 0.2 a | 0.2 ± 0.1 a |

Results are expressed as means ± standard deviations of three independent samples, each with three replicates. Different letters in the same row indicate significant differences ($p < 0.05$).

The diversity of raw materials with which fermented beverages are prepared allows the content of proteins, ash, and soluble solids to be variable. The highest percentages of protein were present in Chicha and Masato, probably due to the quantity of maize used. Depending on the variety of maize, protein can range from 6 to 12% [31].

The high saccharose content in Champús and Masato relates to adding panela (sugar cane) to sweetened beverages. Fructose was significantly detected in Champús due to adding fruits (21.1 ± 14 g/L). Glucose was detected in both Forcha and Champús (around 24 g/L).

Anti-Radical Capacity and Total Phenol Index

The DPPH radical quenching activity of the four different beverages is shown in Table 4. As observed, the radical scavenging activity was higher in Chicha (15.24 ± 3.76 μM TE/L) than in the other beverages, with relatively lower quenching capacity (ranging from 1.99 to 3.08 μM TE/L).

Results of ABTS radical scavenging assays on the maize beverages revealed that Chicha and Champús were effective anti-radical agents; in particular, Chicha showed 15.48 ± 1.05 μM TE/L and Champús showed 13.13 ± 9.19 μM TE/L. On the contrary, the values of Forcha and Masato were shallow at 0.55–0.38 μM TE/L, respectively. The primary content of the total phenol index (TPI) in Chicha could be due to the pre-fermentation process, the quantity of maize used to make the beverage, and the microbial activity. To

the best of our knowledge, this is the first time reporting these measurements in this kind of beverage.

**Table 4.** Antioxidant capacity (DPPH and ABTS) and total phenol index (TPI) of Colombian fermented beverages.

| Beverage | DPPH (μM TE/L) | ABTS (μM TE/L) | TPI (mg GAE/L) |
|---|---|---|---|
| Chicha | 15.24 ± 3.76 b | 15.48 ± 1.05 b | 546 ± 33 b |
| Forcha | 3.08 ± 1.28 a | 0.55 ± 0.42 a | 0.87 ± 0.33 ac |
| Champús | 1.99 ± 0.45 a | 13.13 ± 9.19 b | 258 ± 76 bc |
| Masato | 2.20 ± 0.89 a | 0.38 ± 0.20 a | 0.43 ± 0.15 a |

TE: Trolox equivalent. Results are expressed as means ± standard deviations of three independent samples, each with three replicates. Different letters in the same column indicate significant differences ($p < 0.05$).

Although there is a good correlation ($r^2 = 0.84$) between TPI and antioxidant activity, it is worthy of note that both beverages are made with yellow maize, which contains a significant amount of carotenoids; in addition, Champús is made with the addition of pineapple and lulo, fruits that contain high quantities of vitamin C, also contributing to the antioxidant activity of the beverage [32].

## 4. Discussion

In this study, four types of fermented maize beverages collected from different parts of Colombia were analysed, and microbial communities were identified using Illumina NGS and metagenomic analysis.

The different maize beverages studied here are processed and fermented, following other regional traditions. As observed in these Colombian beverages, fermentation is performed mainly by LAB and yeasts, which contribute to product conservation. These Colombian beverages are still manufactured at home, where the incubation temperature is not easily controlled; in some regions, there is a high oscillation between day and night. In these conditions, the environment exerts a selective pressure that consolidates the dominance of selected species during fermentation [33]. In addition, the influence of the process variables and their variations during the continuous propagation leads to a microbial association that is stable over time [8].

It is well known that the natural fermentation of Colombian fermented maize is characterized by a succession of microorganisms in which the predominant microbial groups are lactic acid bacteria (LAB) and yeasts [8]. A large diversity of LAB species have been shown to have high amylolytic activity. Thus, they have an ecological advantage in fermented maize products as they can partially hydrolyze raw starch to provide sugars such as glucose or maltose that can be used as an energy source by other microorganisms. In contrast, the role of yeasts in these products needs additional research to evaluate their possible impact on starch degradation, aroma, and free amino acid production [8].

Bacteria and yeast populations in fermented beverages interact via multiple mechanisms. The effects of such interactions on the fitness of the single strains may be either positive, neutral, or negative [34]. These interactions could be modified by adding some ingredients (like fruits in the case of Champús), significantly affecting the beverage flavour, rheology, and shelf-life, as well as the chemical characteristics as evidenced by their compositional analysis.

Herein, the metagenomic analysis of the four Colombian maize-fermented beverages pointed out that a complex microbiota is associated with these products and that community membership and structure significantly differed depending on the production process. Bacteria and yeast composition varied in terms of species and their relative abundance. In particular, Forcha was dominated by LAB, which accounts for more than 53% of reads, while yeasts dominated Champús and Chicha. In the Masato beverage, the reads of bacteria were similar to those of yeasts. From the present findings, we also speculated that the

processing, environments, water quality, and storage conditions could have also influenced the microbiota composition in the different fermented maize beverages [35,36].

In general, the dominance of LAB is of great importance in beverages since they can inhibit harmful microorganisms and maintain food safety by lowering the pH or by the presence of bactericidal or fungicidal compounds [37]. In our case, the LAB group was represented by the species belonging to the *Lactobacillus, Lactococcus,* and *Leuconostoc* genus. However, the bacteria composition differed at the species level depending on each local product. In particular, except for Champús, *Lactobacillus* exhibited the highest abundance (between 70% and 89.5%) in the other fermented maize beverages. It is well known that *Lactobacillus* spp. plays an essential role in several fermented foods, and their abundance is related to lactic acid production, which is essential to reduce the pH, as well as to the production of bacteriocins which contribute to the inhibition of pathogenic microorganisms [37]. Except for *L. ruminis, L. sakei, L. sanfranciscensis, L. kefiranofaciens, L. gelidum, L. citreum,* and *O. oeni,* many LAB species reported here have been frequently associated with other fermented maize products worldwide [15,38]. In this regard, *L. sakei* has been isolated from various fermented products such as sourdough, sausages, sauerkraut, Kimchi and Moto [39]. *L. sanfranciscensis,* which prefers maltose as a carbon source, can produce exopolysaccharides (EPS) and phytases. This species has been frequently isolated from sourdough [40] fruit flies and grain beetles [41]. *L. ruminis* is frequently isolated from the gut microbiota, *L. gelidum* and *L. citreum are* isolated from Kimchi [42], and *O. oeni* is a predominant species in wine, cider, and kombucha [43].

Some LAB possess amylolytic activity, enabling them to use maize starch as an energy source. In this context, of particular interest was the presence of *L. fermentum,* a specie that was present in all the fermented beverages, even if in a moderate relative abundance. This specie, together with *L. plantarum,* were the most frequent microorganisms isolated from African fermented maize based [4,44]. The importance of amylolytic LAB during maize fermentation lies in the fact that they may assist in accelerating the rate of acidification by increasing the availability of energy sources, such as glucose or maltose. These sugars can be used by different microorganisms. Amylolytic lactic acid bacteria may also favourably modify the fermented products' rheology [45]. Previous research [27] has suggested that amylolytic activity, acid tolerance, and bacteriocin production contributed to the consolidated presence of *L. fermentum* in fermented maize foods.

In addition to *L. fermentum,* other identified OTUs were frequently found. However, their presence varied depending on the beverage, and this was the case of *L. plantarum, L. reuteri, L. ruminis, L. buchneri, L. lactis, Leuconostoc mesenteroides,* and *W. koreensis,* which were found in at least 3 out 4 beverages here studied. These species have been related to beneficial effects on health; in this sense, *L. plantarum* significantly decreased LDL-cholesterol and apolipoprotein (Apo)B-100 [46]. In addition, it has been reported that *L. fermentum* intakes reduced intestinal inflammation by inhibiting inflammatory mediators and the infection of pathogenic species by increasing intestinal barrier function (by increasing tight junction proteins) [47]. On the other hand, *L. lactis, L. citrinum,* and *L. mesenteroides,* present in Forcha and Chicha, have been reported to produce bacteriocins [48–50]. Studies by Yeong et al. (2020) using *W. koreensis* isolated from Kimchi revealed that this specie produced ornithine from arginine through the arginine deiminase (ADI) and ornithine transcarbamoylase pathway; this amino acid is related to stress reduction, collagen synthesis, muscle growth and increased basal metabolism [51]. In addition, *L. mensenteroides* uses maize starch to produce non-digestible oligosaccharides and extracellular polysaccharides, such as dextran, which are highly appreciated for their prebiotics properties [52].

An interesting result is the high relative abundance of the *O. oeni* in Chicha and Forcha beverages, which require more restrictive growth conditions than another LAB. This characteristic is indicated by their acidophilic nature, adaptation to the wine environment, alcohol tolerance, and requirement for specific growth factors [43]. To the best of our knowledge, this is the first time that *O. oeni* has been reported in fermented maize beverages. Furthermore, a high relative abundance of *W. koreensis* was found mainly in Champús and

Chicha; this specie is a strictly heterofermentative genus, producing lactic acid and acetic acid as the final products of the carbohydrate metabolism [53]. The frequent detection of *Weissella* spp. in a large variety of spontaneously fermented foods proves that they can adapt to many environments and play an essential role in the fermentation process [54]. Regarding bacteriocins producers LAB, we detected the presence of several species reported as bacteriocin producers; among the species caught, *L. plantarum* produces plantaricin and can be adapted to various niches thanks to its ability to ferment a wide range of carbohydrates, including *L. helveticus* (Lactacin-Helveticin), *Lactobacillus delbrueckii* spp. bulgaricus (Bulgaricin), *L. brevis* (Lactobacillin), *L. fermentum* (Bacteriocin), *Lactobacillus acidophilus* (Lactacin) and *L. sakei* (Sakacin) [55]. In addition, we found *L. mesenteroides*, another LAB specie that produces the bacteriocin mesenterocin.

Our study also revealed that *A. pasteurianus* was present in all the beverages, although their relative abundance was low. This specie, together with *Komagataeibacter medellinensis* (*Gluconacetobacter medellinensis*) and *Gluconobacter oxydans*, have been reported in fermented drinks based on cereals, such as sour Atole and sourdough [15], this last specie of which was also present during the fermentation of coffee and cocoa beans [56,57]. This group of bacteria is characterized by the ability to oxidize carbohydrates, sugars, alcohols, and/or ethanol into organic acids such as acetic acid, a compound considered antimicrobial and antioxidant [58], and for energy production through a specific respiratory chain [59]. *G. oxydans* mainly produce L-sorbose from D-sorbitol and 2,5-diketogluconic from glucose, making it a species of biotechnological importance in the production of vitamin C [60].

In the case of yeast, they play an essential role in fermented beverages. In fact, besides fermenting carbohydrates, they contribute to flavour compound formation, stimulate the growth of lactic acid bacteria, and produce different enzymes able to degrade cyanogenic glycosides. They are also reported to bind mycotoxins and have probiotic properties [8]. Although the metanalyses did not show any yeast species whose presence was familiar in all the beverages, *S. cerevisiae*, which is mainly responsible for ethanol production and $CO_2$, is one of the main constituents of the stable microbial population in Forcha, Chicha, and Masato. Although *S. cerevisiae* was prevalent in Chicha and Forcha, there was a big difference in ethanol content (Table 3). Two reasons may explain this. Firstly, DNA sequencing provides a snapshot of the bacterial and yeast population; however, it may not be able to distinguish between non-metabolically and metabolically active cells. Thus, it could be probable that a significant part of *S. cerevisiae* in Forcha was not active. Secondly, *S. cerevisiae* could have gradually shifted the fermentative metabolism towards a respiratory phenotype accompanied by a decrease in growth and glucose uptake rate, as can occur with the aging of the cells [61]. This specie is also the predominant yeast species from fermented maize beverages in Africa [62] and Andean countries such as Argentina [63,64], Ecuador [65,66], and Perú [52]. As evidenced by the Shannon index, Champús was the beverage with the most significant diversity in yeast, probably due to the addition of fruits. The alcohol-tolerant non-Saccharomyces yeasts such as *H. nectarophila*, *S. stellimalicola*, *P. eremophila*, and *P. kudriavzevii* prevail among the predominant species in this type of beverage. In particular, *H. nectarophila* was isolated for the first time from a flower of *Siphocampylus corymbiferus* in the Atlantic rainforest of São Paulo (Brazil) and presented as a new specie in 2014 [67]. It was frequently isolated in Cyprus driven by the grape fermentation together with *S. cerevisiae* [68].

On the other hand, *S. stellimalicola* has been isolated from Dadih buffalo-fermented milk produced in Sumatera [69], and showed antagonist effects against *Penicillium italicum*, which was attributed to the strong chitinase production and the conidial germination inhibition [70]. *P. kluyveri* (frequently used in winemaking to improve aroma) and *P. kudriavzevii* have often been reported in cereal fermented beverages from Africa and South America [14,71]. They have also been reported to produce significant amounts of the enzymes phytases important to degrade phytic acid, an anti-nutritional factor in maize. Another yeast well known for its high acid- and ethanol-resistance is *D. bruxellensis*, the sexually reproducing form of *Brettanomyces bruxellensis*, which was present in Forcha, and Masato

was found in Mahewu and Umqombothi, fermented beverages from South Africa both made from a combination of maize and sorghum [72], but also in other beers and wine. The presence of this specie in fermented beverages is often ambiguous because some beers give desirable aromas while different beer styles and wines confer unpleasant aromas [73].

It is well known that fermentation improves the nutritional, functional, and safety of maize products and influences their antioxidant properties, contributing to the health-promoting usage of these products. In particular, yellow maize contains a significant amount of carotenoids that increase during fermentation [74], and phenolic compounds in conjugated forms are released to free phenolics to enhance their potency [75]. In this context, Svensson et al. 2009 [76] reported that during lactic fermentation of sorghum using *L. plantarum* and *L. casei* or *L. fermentum* and *L. reuteri* in single or binary phenolic acids, phenolic acid esters and flavonoid glucosides were metabolized. Gänzle 2014 [77] highlights that reductases and decarboxylases mediate phenolic acid metabolism in some strain LAB as a mechanism of detoxification. In our research, we observed a high TPI, probably due to the addition of the lulo (*Solanum quitoense* Lam.), a fruit known as naranjilla. This fruit contains all-trans–carotene, 13-cis-β-carotene, and 9-cis-β-carotene and the lutein (carotenoids); chlorogenic acids and their hexosides in the flesh and placental tissues, and flavonol glycosides in the skin (phenolic compounds) [78]. Phenolic compounds are reported in other fermented beverages, such as beer and wine. Their moderate intake has been associated with decreased lipid peroxidation levels and a lower incidence of certain types of cancer [75,79], indicating the functional properties of these beverages.

## 5. Conclusions

In our study, we unveiled the structure of the bacterial and yeast community of the most popular maize-fermented beverages from Colombia with high-throughput multiplex barcoded pyrosequencing. The different bacteria and yeast species found have been reported to have the metabolic capacity to perform specific functional roles during fermentation, playing a vital role in the quality, taste and aroma of each fermented beverage. Only two species, *L. fermentum* and *A. pasteurianus,* were identified to be the most persistent in all types of fermented beverages. However, they were not the most abundant, thus underlying the capability of these inhabitant bacteria to adjust to the constantly changing fermentation environment. To the best of our knowledge, this is the first report to analyze the core microbiota in these beverages. In addition, the unique characteristics found in the traditional fermented beverages here studied, such as the antioxidant capacity, physicochemical characteristics, and their significant nutritional and functional contribution, depend mainly on the native raw materials of the region and the utensils where they are elaborated, and not only in the elaboration process. This is of vital importance to scale the process for massive consumption.

**Supplementary Materials:** The following are available online at https://www.mdpi.com/article/10.3390/fermentation8120733/s1, Table S1: Relative abundance of identified bacterial communities' diversity in Forcha at the Phyla level. Table S2: Relative abundance of identified bacterial communities' diversity in fermented beverages at the genus level. Table S3: Relative abundance of identified bacterial communities' diversity in fermented beverages at the specie level. Table S4: Relative abundance of identified fungal communities' diversity in fermented beverages at the genus level. Table S5: Relative abundance of identified fungal communities' diversity in fermented beverages at the specie level.

**Author Contributions:** Conceptualization, C.D.G.-T., L.F.P.-P., J.D.-O. and C.C.-L.; methodology, L.F.P.-P., L.G.P.-P., J.D.-O. and F.F.F.-D.; validation, J.D.-O., R.A.C. and C.C.-L.; formal analysis, J.D.-O., R.A.C. and C.C.-L.; investigation, L.F.P.-P., D.P.N.-P., C.D.G.-T. and C.C.-L.; resources, L.F.P.-P., C.D.G.-T. and J.D.-O.; data curation, F.F.F.-D. and R.A.C.; writing—original draft preparation, L.F.P.-P., J.D.-O., R.A.C. and C.C.-L.; writing—review and editing, J.D.-O. and C.C.-L.; project administration, C.D.G.-T. and J.D.-O.; funding acquisition, J.D.-O., D.P.N.-P. and C.D.G.-T. All authors have read and agreed to the published version of the manuscript.

**Funding:** The authors acknowledge the support provided by Universidad de San Buenaventura Cali (USB-2018), University of Teramo (Unite-2019), and Universidad del Atlántico (UA-2022).

**Institutional Review Board Statement:** Not applicable.

**Data Availability Statement:** The data presented in this study are available on request from the corresponding author.

**Conflicts of Interest:** The authors declare that they have no conflict of interest.

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
