# Peer review of "Exploring the Core Microbiota of Four Different Traditional Fermented Beverages from the Colombian Andes"

_fermentation, doi:10.3390/fermentation8120733_

Round 1

Reviewer 1 Report

Comments on fermentation-2060597

This manuscript titled “Exploring the core microbiota of four different traditional fermented beverages from the Colombian Andes” (fermentation 2060597) investigated the dominant bacterial and fungal community responsible for the fermentation of four indigenous beverages via metagenomics analysis. Authors also described the antioxidant activity of the four beverages. It is an interesting study for exploring rarer bacterial and fungal genera for the traditional drinks. The experiments in this study are coherent, and the results are well presented. However, some problems have to be addressed for authors’ consideration before resubmitting to this Journal.

1.     Line 61-72, references related to the processing of the four drinks are necessary.

2.     Line 85 for “the traditional fermented beverage samples”, could provide more information for the samples, for instance, shelf-life, chemical composition or even photos?

3.     Line 137. Why were the three saccharides (saccharose, glucose, fructose) and only lactic acid selected (because the acetic bacteria also existed, see line 193)?

4.     Line 205-206, please clarify “between 260053 and 338532 for bacteria and 297640 and 405662 for fungi”.

5.     Line 345-358. Reference should be supplemented to support the conjecture. As one of alternative, major fermentation substrates should be listed for each of the four beverages.

6.     Line 359 for table 3, there is big difference in ethanol level between the Chicha (9.9) and Forcha (2.9) samples. However, S. cerevisiae as the most common yeast responsible for alcohol formation were 77.6% for Forcha, and 84.79% for Chicha (line274-274), and thus no big difference for them was observed. How to explain this point, depending on processing, raw materials (see line 528-529) or other factors?

7.     Line 409. No evidence supports this view that “processing, environments, water quality, and storage conditions” influence the microbiota composition. Moreover, authors should make further discussion about the relationship between bacteria/yeast composition and fermentation time, and the relationship between bacteria/yeast composition and fermentation substrates.

8.     As the authors reported, “beverages have been prepared for at home or in the cottage industry using simple techniques and equipment”. How do the authors ensure that each sample is representative?

9.     Line 534-535, this sentence means a functionality when a long consumption of the four beverages (see line 546-551)?

10. For the part Reference, the Latin name of each species should be spelled in italic, and their format should be unified. References should be replaced by recent studies such as 17, 18, 19, and 40.

Author Response

Dear Reviewer 1,

Thank you for your helpful comments and for pointing out options to improve our manuscript. We modified the paper according to your suggestions.

This manuscript titled "Exploring the core microbiota of four different traditional fermented beverages from the Colombian Andes" (fermentation 2060597) investigated the dominant bacterial and fungal community responsible for the fermentation of four indigenous beverages via metagenomics analysis. The authors also described the antioxidant activity of the four beverages. It is an interesting study exploring rarer bacterial and fungal genera for traditional drinks. The experiments in this study are coherent, and the results are well presented. However, some problems have to be addressed for authors' consideration before resubmitting to this Journal.

  1. Line 61-72, references related to the processing of the four drinks are necessary.

Reply: References related to the beverage process were included. Colombia's research on these beverages is still unexplored (Lines 68-77).

  1. Chaves-López, C.; Serio, A.; Grande-Tovar, C.D.; Cuervo-Mulet, R.; Delgado-Ospina, J.; Paparella, A. Traditional fermented foods and beverages from a microbiological and nutritional perspective: The Colombian Heritage. Compr. Rev. Food Sci. Food Saf. 2014, 13, doi:10.1111/1541-4337.12098.
  2. My colombian cocina. Available online: https://www.mycolombiancocina.com/ (accessed on Dec 1, 2022).
  3. Quintero-Angel, M.; Mendoza-Salazar, D.M.; Martínez-Girón, J. Food fears and risk of loss of food heritage: A little-explored effect of food modernity and times of pandemic. Int. J. Gastron. Food Sci. 2022, 28, 100499, doi:10.1016/j.ijgfs.2022.100499.
  4. Osorio-Cadavid, E.; Chaves-López, C.; Tofalo, R.; Paparella, A.; Suzzi, G. Detection and identification of wild yeasts in Champús, a fermented Colombian maize beverage. Food Microbiol. 2008, 25, 771–777, doi:10.1016/j.fm.2008.04.014.

  1. Line 85 for "the traditional fermented beverage samples" could provide more information for the samples, for instance, shelf-life, chemical composition or even photos?

Reply: Figure 1 was included in the introduction with the information on the preparation of the beverages (Line 67). Additionally, the sentence was included. Lines 92-94: Under refrigerated conditions, these beverages remain acceptable for one week, except for Chicha, which remains acceptable for about two weeks.

Figure 1. Flowchart of the production of four traditional fermented beverages from the Colombian Andes (Chicha, Forcha, Champús, and Masato) elaborated from maize.

  1. Line 137. Why were the three saccharides (saccharose, glucose, fructose) and only lactic acid selected (because the acetic bacteria also existed, see line 193)?

Reply: Our analyses detected only very few quantities that do not characterize the beverages. For this reason, we did not include these data.

  1. Line 205-206, please clarify "between 260053 and 338532 for bacteria and 297640 and 405662 for fungi".

Reply: The sentence has been rearranged for clarity.

Line 222-228. The proportions of high-quality sequences were above 79.7% in all beverages. The number of high-quality sequences ranged between 260053 and 338532 for bacteria and 297640 and 405662 for fungi, indicating that the sequences obtained were sufficient to represent the microbial structure of the beverage samples. Bacteria comprised 79.76%, 86.23%, 85.40%, and 86.14% of assigned reads across the different beverages (Masato, Forcha, Champús, and Chicha, respectively), and fungi made up 84.29%, 82.94%, 87.87%, and 85.46% of assigned reads.

  1. Line 345-358. Reference should be supplemented to support the conjecture.

Reply: Thanks for your suggestion; in line 371, we added: Depending on the variety of maize, protein can range from 8 to 12% (Vargas-Escobar et al. 2016).

  1. Vargas Escobar, E.A.; Baena García, D.; Vargas Sánchez, J.E. Análisis de estabilidad y adaptabilidad de híbridos de maíz de alta calidad proteica en diferentes zonas agroecológicas de Colombia. Acta Agronómica 2016, 65, 72–79, doi:10.15446/acag.v65n1.43417.

As one of alternative, major fermentation substrates should be listed for each of the four beverages.

Reply: In figure 1, we include the major fermented substrates.

  1. Line 359 for table 3, there is big difference in ethanol level between the Chicha (9.9) and Forcha (2.9) samples. However, S. cerevisiae as the most common yeast responsible for alcohol formation were 77.6% for Forcha, and 84.79% for Chicha (line274-274), and thus no big difference for them was observed. How to explain this point, depending on processing, raw materials (see line 528-529) or other factors?

Reply:  Thanks for this observation. In lines 518-525, we state that: Although S. cerevisiae was prevalent in Chicha and Forcha, there was a big difference in ethanol content (Table 3). Two reasons may explain this. Firstly, DNA sequencing provides a snapshot of the bacterial and yeast population; however, it may not be able to distinguish between non-metabolically and metabolically active cells. Thus, a significant part of S. cerevisiae in Forcha could probably not be active. Secondly, S. cerevisiae could have gradually shifted the fermentative metabolism towards a respiratory phenotype which is accompanied by a decrease in growth and glucose uptake rate as can occurs with the age of the cells (Leupold, S.; Hubmann, G.; Litsios, A.; Meinema, A.C.; Takhaveev, V.; Papagiannakis, A.; Niebel, B.; Janssens, G.; Siegel, D.; Heinemann, M. Saccharomyces cerevisiae goes through distinct metabolic phases during its replicative lifespan. Elife 2019, 8, e41046, doi:10.7554/eLife.41046.).

  1. Line 409. No evidence supports this view that "processing, environments, water quality, and storage conditions" influence the microbiota composition.

Reply: in line 436, we added two references in which the authors studied the influence of some ecological factors on the microbiota of fermented cereals.

  1. Minervini, F.; De Angelis, M.; Di Cagno, R.; Gobbetti, M. Ecological parameters influencing microbial diversity and stability of traditional sourdough. Int. J. Food Microbiol. 2014, 171, 136–146, doi:10.1016/j.ijfoodmicro.2013.11.021.
  2. Minervini, F.; Dinardo, F.R.; De Angelis, M.; Gobbetti, M. Tap water is one of the drivers that establish and assembly the lactic acid bacterium biota during sourdough preparation. Sci. Rep. 2019, 9, 570, doi:10.1038/s41598-018-36786-2.

Moreover, authors should make further discussion about the relationship between bacteria/yeast composition and fermentation time, and the relationship between bacteria/yeast composition and fermentation substrates.

Reply: In lines 414-421, we stated that: It is well known that the Natural fermentation of Colombian fermented maize is characterized by a succession of microorganisms in which the predominant microbial groups are lactic acid bacteria (LAB) and yeasts [8]. A large diversity of LAB species have been shown to have high amylolytic activity. Thus, they have an ecological advantage in fermented maize products as they can partially hydrolyse raw starch to provide sugars such as glucose or maltose that can be used as an energy source by other microorganisms. In contrast, the role of yeasts in these products needs additional research to evaluate their possible impact on starch degradation, aroma, and free amino acid production [8].

  1. Chaves-López, C.; Serio, A.; Grande-Tovar, C.D.; Cuervo-Mulet, R.; Delgado-Ospina, J.; Paparella, A. Traditional fermented foods and beverages from a microbiological and nutritional perspective: The Colombian Heritage. Compr. Rev. Food Sci. Food Saf. 2014, 13, doi:10.1111/1541-4337.12098.

  1. As the authors reported, "beverages have been prepared for at home or in the cottage industry using simple techniques and equipment". How do the authors ensure that each sample is representative?

Reply: This work considers the most knowledgeable and traditional local producers characterized by natural fermentation. These beverages were recognized as better beverages in the locality.

  1. Line 534-535, this sentence means a functionality when a long consumption of the four beverages (see line 546-551)?

Reply: in lines 565-566, we state, "indicating the functional properties of these beverages.

  1. For the part Reference, the Latin name of each species should be spelled in italic, and their format should be unified. References should be replaced by recent studies such as 17, 18, 19, and 40

Reply: references were updated by recent studies.

  1. Ilyasov, I.R.; Beloborodov, V.L.; Selivanova, I.A.; Terekhov, R.P. ABTS/PP Decolorization Assay of Antioxidant Capacity Reaction Pathways. Int. J. Mol. Sci. 2020, 21, 1131, doi:10.3390/ijms21031131.
  2. Villa-Rodríguez, J.A.; Molina-Corral, F.J.; Ayala-Zavala, J.F.; Olivas, G.I.; González-Aguilar, G.A. Effect of maturity stage on the content of fatty acids and antioxidant activity of "Hass" avocado. Food Res. Int. 2011, 44, 1231–1237, doi:10.1016/j.foodres.2010.11.012.
  3. Ioannone, F.; Di Mattia, C.D.; De Gregorio, M.; Sergi, M.; Serafini, M.; Sacchetti, G. Flavanols, proanthocyanidins and antioxidant activity changes during cocoa (Theobroma cacao L.) roasting as affected by temperature and time of processing. Food Chem. 2015, 174, 263–269, doi:10.1016/j.foodchem.2014.11.019.
  4. Lau, S.W.; Chong, A.Q.; Chin, N.L.; Talib, R.A.; Basha, R.K. Sourdough Microbiome Comparison and Benefits. Microorganisms 2021, 9, 1355, doi:10.3390/microorganisms9071355.
  5. Maske, B.L.; de Melo Pereira, G. V; da S. Vale, A.; de Carvalho Neto, D.P.; Karp, S.G.; Viesser, J.A.; De Dea Lindner, J.; Pagnoncelli, M.G.; Soccol, V.T.; Soccol, C.R. A review on enzyme-producing lactobacilli associated with the human digestive process: From metabolism to application. Enzyme Microb. Technol. 2021, 149, 109836, doi:10.1016/j.enzmictec.2021.109836.
  6. Azhar, N.S.; Md Zin, N.H.; Hamid, T. Lactococcus lactis strain A5 producing nisin-like bacteriocin active against Gram positive and negative bacteria. Trop. life Sci. Res. 2017, 28, 107–118, doi:10.21315/tlsr2017.28.2.8.

Reviewer 2 Report

The manuscript deals with the analysis of the core microbiota of four different traditional fermented beverages from the Colombian Andes.

The paper is well written, is innovative and provides valuable information dealing with the core microbiota of the four beverages using metagenomic sequencing.

However, I have some questions raised when I was reading this study that should be appropriately addressed by the authors.

The Material and Methods section is incomplete and should be rewritten to provide more information about the analytical and statistical methods used in the manuscript. In addition, the results section should also be rewritten based on the changes made in the Material and Methods section.

I strongly recommended that all authors carefully revise the manuscript to correct the methodological errors detected in the manuscript. There are sufficient authors to make an appropriate and efficient revision of the paper.

Other considerations are as follows:

2. Materials and methods section.

2.1. Line 88: Please replace “4°C” with “4 °C”. Please separate the number and the unit. This is repeated throughout the manuscript.

2.2. Lines 90100:  How many samples did the authors used for physicochemical characterization? The same for all analytical and microbiological methods.

2.3. Line 91: Please replace “25°C” with “25 °C”.

2.4. Line 91: It is not necessary to say that the pH value was measured with a pH meter. So that, this sentence should be deleted.

2.5. Line 93: Please replace “1 minute” with “1 min.”

2.6. Line 95: Please replace “24 hours” with “24 h”. Please replace “65°C” with “65 °C”.

2.7. Line 96: Please replace “36 hours” with “36 h”. Please replace “600°C” with “600 °C”.

2.8. Line 99: Please replace “2h” with “2 h”. Please replace “420°C” with “420 °C”.

2.9. Line 104: Please replace “25°C” with “25 °C”.

2.10. Line 112: Please replace “6 minutes” with “6 min.”

2.11. Lines 123131: The method Folin-Ciocalteu is not a suitable method for measuring total phenolic content due to the interference of protein compounds, organic acids and reducing sugars present in the beverages with Folin's reagent. This could lead to an overestimation of the total phenolic content. For this reason, it would more appropriate to use a HPLC method to measure total phenolic content in the beverages (Pérez-Gregorio et al., 2011). Please provide a suitable explanation for this fact.

2.12. Lines 133 and 140: Please replace “H2SO4 5 mM” with “5 mM H2SO4

2.13. Lines 143–152: The authors used the MRS, GYC and YPD agar as a culture medium for the incubation of lactic acid bacteria (LAB), acetic acid bacteria (AAB) and yeasts, respectively. However, recent studies showed that the addition of an antifungal compound (e.g. amphotericin B or nystatin) to the MRS and GYC agar media, after media sterilization, is necessary to prevent fungal growth (Gao et al., 2018; Bazán et al., 2022). In the same way, the addition of an antibacterial compound (e.g. chloramphenicol) to YEG agar after medium sterilization is necessary to prevent bacterial growth (Corona et al., 2016; Randazzo et al., 2016; van Eldere et al., 1996; Gao et al., 2018; Bazán et al., 2022; Moretti et al., 2022). However, the authors used the media for incubation and count of the different microbial groups, but they do not supplemented the media with amphotericin B or nystatin (in the case of MRS and GYC agar media) of chloramphenicol (in the case of YPD agar medium). So that, yeasts and could grow in MRS and GYC agar media, or LAB and AAB could grow in YPD agar medium, leading to erroneous counts of each microbial group. The authors should provide an explanation to support the use of the MRS, GYC and YPD agar without these supplements.

2.14. Lines 184–190: Commonly, the experimental data are standardized by transforming the original value of each variable into a z score, before clustering:

zi = (yi-ȳ)/SDy 

where zi is the z score, yi is the original value of each variable, ȳ is the mean of all values of y, and SDy is the standard deviation of that mean.

With this transformation, each variable, which has a mean of 0 and a standard deviation of 1, and these variables can equally contribute to the discrimination process, being homogenized both the magnitude and variability of all mean concentrations of each variable (Alonso et al., 2010). Did the authors standardize the variables used in the discrimination process before clustering? How did they do it?  (Alonso et al., 2010).

Which variable was used as the distance measure or similarity index? What amalgamation (linkage) method was used to build the dendograms? These informations should be included in the Materials and Methods section.

2.15. Lines 194 and Table 1: How did the authors detected the presence of coliforms in the beverages? This method was not explained in the Materials and methods section. Eight authors sign the paper and surprisingly, none seemed to detect this error. I strongly recommend that all authors carefully read the manuscript to avoid these errors.

2.16. Lines 244–251: The genus of the microorganism should be written in abbreviate form after being mentioned for the first time. For example, “On the contrary, in Masato Lactobacillus fermentum, Lactobacillus ruminis, Lactobacillus reuteri and Acetobacter pasteurianus were predominant. In addition, in Champús, Lactobacillus fermentum, Weisella koreensis, Lactobacillus buchneri, Lactobacillus brevis, Lactobacillus reuteri, and Acetobacter pasteurianus were significantly (p < 0.05) abundant while in Chicha were Lactobacillus lactis and Weissella koreensis” should be replaced with “On the contrary, in Masato Lactobacillus fermentum, L. ruminis, L. reuteri and Acetobacter pasteurianus were predominant. In addition, in Champús, L. fermentum, Weisella koreensis, L. buchneri, L. brevis, L. reuteri, and A. pasteurianus were significantly (p < 0.05) abundant while in Chicha were L. lactis and W. koreensis

2.17. Table 3: Why did not the authors statistically compare the data of the chemical composition of the four fermented beverages?

2.18. Table 4: Why did not the authors statistically compare the data of the antioxidant capacity (DPPH and ABTS), total phenolic content (TPC) of the four Colombian fermented beverages?

The above-mentioned methodological errors (in the materials and methods section and Results section) should be appropriately addressed to provide a sounder and deeper discussion of the results to give appropriate conclusions supported by a rigorous analysis of the results obtained.

References used in this letter:

Alonso et al. Production and characterization of distilled alcoholic beverages obtained by solid-state fermentation of black mulberry (Morus nigra L.) and black currant (Ribes nigrum L.). J. Agric. Food Chem. 2010, 58, 2529–2535.

van Eldere et al. Fluconazole and amphotericin B antifungal susceptibility testing by National Committee for Clinical Laboratory Standards broth macrodilution method compared with E-test and semiautomated broth microdilution test. J. Clin. Microbiol. 1996, 34, 842847.

Corona et al. Characterization of kefir-like beverages produced from vegetable juices. LWT‐Food Sci. Technol. 2016, 66, 572−581.

Randazzo et al. Development of new non‐dairy beverages from Mediterranean fruit juices fermented with water kefir microorganisms. Food Microbiol. 2016, 54, 40–51.

Bazán et al. The chemical, microbiological and volatile composition of kefir‐like beverages produced from red table grape juice in repeated 24‐h fed‐batch subcultures. Foods. 2022, 11, 3117.

Moretti et al. Water kefir, a fermented beverage containing probiotic microorganisms: From ancient and artisanal manufacture to industrialized and regulated commercialization. Future Foods. 2022, 5, 100123.

Gao et al. Microbiological characterisation of whey-based kefir beverages after Bod ljong cheese-making at different fermentation temperatures. IOP Conf. Series: Materials Science and Engineering. 2018, 392, 052010.

Pérez-Gregorio et al. Influence of alcoholic fermentation process on antioxidant activity and phenolic levels from mulberries (Morus nigra L.). LWT‐Food Sci. Technol. 2011, 44, 1793–1801.

Author Response

Dear Reviewer 2,

Thank you for your helpful comments and for pointing out options to improve our manuscript. We modified the paper according to your suggestions.

The manuscript deals with the analysis of the core microbiota of four different traditional fermented beverages from the Colombian Andes.

The paper is well written, is innovative and provides valuable information dealing with the core microbiota of the four beverages using metagenomic sequencing.

However, I have some questions raised when I was reading this study that should be appropriately addressed by the authors.

The Material and Methods section is incomplete and should be rewritten to provide more information about the analytical and statistical methods used in the manuscript. In addition, the results section should also be rewritten based on the changes made in the Material and Methods section.

I strongly recommended that all authors carefully revise the manuscript to correct the methodological errors detected in the manuscript. There are sufficient authors to make an appropriate and efficient revision of the paper.

Reply: We have accepted your suggestion and modified the Material and Methods section

Other considerations are as follows:

  1. Materials and methods section.

2.1. Line 88: Please replace "4°C" with "4 °C". Please separate the number and the unit. This is repeated throughout the manuscript.

It was changed.

2.2. Lines 90–100:  How many samples did the authors used for physicochemical characterization? The same for all analytical and microbiological methods.

Reply. We are very grateful for this comment. In section 2.1, the number of samples was reported. Emphasis will be placed on each analytical method (Lines 105-106; 113-114; 142-143; 165).

2.3. Line 91: Please replace "25°C" with "25 °C".

It was changed.

2.4. Line 91: It is not necessary to say that the pH value was measured with a pH meter. So that, this sentence should be deleted.

It was changed.

2.5. Line 93: Please replace "1 minute" with "1 min."

It was changed.

2.6. Line 95: Please replace "24 hours" with "24 h". Please replace "65°C" with "65 °C".

It was changed.

2.7. Line 96: Please replace "36 hours" with "36 h". Please replace "600°C" with "600 °C".

It was changed.

2.8. Line 99: Please replace "2h" with "2 h". Please replace "420°C" with "420 °C".

It was changed.

2.9. Line 104: Please replace "25°C" with "25 °C".

It was changed.

2.10. Line 112: Please replace "6 minutes" with "6 min."

It was changed.

2.11. Lines 123–131: The method Folin-Ciocalteu is not suitable for measuring total phenolic content due to the interference of protein compounds, organic acids and reducing sugars present in the beverages with Folin's reagent. This could lead to an overestimation of the total phenolic content. For this reason, it would be more appropriate to use an HPLC method to measure the total phenolic content in the beverages (Pérez-Gregorio et al., 2011). Please provide a suitable explanation for this fact.

Reply. Undoubtedly, the most popular and widespread method to measure phenolics is their titration with the Folin-Ciocalteu's reagent. This reagent was used to titrate phenolic compounds in wine, but it has been extensively used to measure phenolic antioxidants in food or vegetable extracts. Its mechanism is based on an oxidation/reduction reaction as polyphenols are oxidized in a primary environment by the Folin-Ciocalteu reagent (FC), a mixture of tungstate and molybdate. The consequence of the reaction is the formation of coloured molybdenum ions, MoO4+, which absorb in the interval 725-765 nm. The method is simple, sensitive, and precise, but the reaction lacks specificity: any reducing compounds, like sugars, aromatic amines, sulfur dioxide, ascorbic acid, Fe(II) and other molecules, can react with the FC reagent; therefore, we understand the reviewer's comment.

However, as this assay relies on a redox reaction, it can also be considered an index of the antioxidant activity or reducing power (Prior et al., 2005). The total phenol index (TPI) has been extensively used as an antioxidant activity index in food science literature.

We changed the 'Total Phenolic Compounds' definition to Total phenol index (TPI), and we discussed that this is an index of reducing power which, in turn, is affected by phenolics and other molecules. We included TPI within the antioxidant activity section and discussion.

2.12. Lines 133 and 140: Please replace "H2SO4 5 mM" with "5 mM H2SO4"

Reply: It was changed.

2.13. Lines 143–152: The authors used the MRS, GYC and YPD agar as a culture medium for incubating lactic acid bacteria (LAB), acetic acid bacteria (AAB) and yeasts, respectively. However, recent studies showed that adding an antifungal compound (e.g. amphotericin B or nystatin) to the MRS and GYC agar media after media sterilization is necessary to prevent fungal growth (Gao et al., 2018; Bazán et al., 2022). In the same way, the addition of an antibacterial compound (e.g. chloramphenicol) to YEG agar after medium sterilization is necessary to prevent bacterial growth (Corona et al., 2016; Randazzo et al., 2016; van Eldere et al., 1996; Gao et al., 2018; Bazán et al., 2022; Moretti et al., 2022). However, the authors used the media for incubation and counted the different microbial groups, but they did not supplement the media with amphotericin B or nystatin (in the case of MRS and GYC agar media) of chloramphenicol (in the case of YPD agar medium). So, yeasts could grow in MRS and GYC agar media, or LAB and AAB could grow in YPD agar medium, leading to erroneous counts of each microbial group. The authors should provide an explanation to support using the MRS, GYC and YPD agar without these supplements.

Reply: Thanks for your observation. It was an overside. In the manuscript, we included the supplements used in these media as follows:

Line 153-164. Lactic Acid Bacteria (LAB) in MRS agar (Oxoid, Basingstoke, UK) supplemented with 250 mg/L of fluconazole (Sigma-Aldrich Italy, Milan, IT) at 37 °C in anaerobiosis for 48 h. Acetic Acid Bacteria (AAB) in GYC agar (20 g/L glucose, 10 g/L yeast extract, 3 g/L CaCO3, 15 g/L agar, 70 g/L ethanol) supplemented with 250 mg/L of fluconazole at 37 °C for 72 h and those that showed halos of degradation were considered positive. Yeasts in YPD agar (Biolife, Italiana, Milan, Italy) were added with 150 ppm chloramphenicol (Sig-ma-Aldrich Italy, Milan, IT) at 25 °C for 24 h, and Molds in DG18 Agar (Oxoid) and Czapec- Agar (Biolife) added with 150 ppm chloramphenicol for 96 h.

2.14. Lines 184–190: Commonly, the experimental data are standardized by transforming the original value of each variable into a z score, before clustering:

zi = (yi-ȳ)/SDy

where zi is the z score, yi is the original value of each variable, ȳ is the mean of all values of y, and SDy is the standard deviation of that mean.

With this transformation, each variable, which has a mean of 0 and a standard deviation of 1, and these variables can equally contribute to the discrimination process, being homogenized both the magnitude and variability of all mean concentrations of each variable (Alonso et al., 2010). Did the authors standardize the variables used in the discrimination process before clustering? How did they do it?  (Alonso et al., 2010).

Which variable was used as the distance measure or similarity index? What amalgamation (linkage) method was used to build the dendograms? These informations should be included in the Materials and Methods section.

Reply: We agree with your comment. This information was included in the Materials and Methods section.

Line 204-207. Data were standardized in percent relationship versus to the number of OTUS in each beverage and transformed into ln(x+1) before clustering. Beverages were clustered using correlation distance and average linkage [26].

2.15. Lines 194 and Table 1: How did the authors detected the presence of coliforms in the beverages? This method was not explained in the Materials and methods section. Eight authors sign the paper and surprisingly, none seemed to detect this error. I strongly recommend that all authors carefully read the manuscript to avoid these errors.

Reply: You are right. For this reason, the method used was incorporated in the materials and methods section and discussed accordingly.

Line 160-164. Total coliforms were isolated in Violet Red Bile Glucose Agar and Violet Red Bile Agar (Oxoid, Basingstoke, UK) at 37°C for 24 hours respectively in anaerobiosis. Coliforms were reconfirmed using EMB agar. The visible colony count at the end of the incubation period and the dilution factor were used to determine the number of microorganisms present in the sample.

2.16. Lines 244–251: The genus of the microorganism should be written in abbreviate form after being mentioned for the first time. For example, "On the contrary, in Masato Lactobacillus fermentum, Lactobacillus ruminis, Lactobacillus reuteri and Acetobacter pasteurianus were predominant. In addition, in Champús, Lactobacillus fermentum, Weisella koreensis, Lactobacillus buchneri, Lactobacillus brevis, Lactobacillus reuteri, and Acetobacter pasteurianus were significantly (p < 0.05) abundant while in Chicha were Lactobacillus lactis and Weissella koreensis" should be replaced with "On the contrary, in Masato Lactobacillus fermentum, L. ruminis, L. reuteri and Acetobacter pasteurianus were predominant. In addition, in Champús, L. fermentum, Weisella koreensis, L. buchneri, L. brevis, L. reuteri, and A. pasteurianus were significantly (p < 0.05) abundant while in Chicha were L. lactis and W. koreensis"

It was revised throughout the document.

2.17. Table 3: Why did not the authors fisictically compare the data of the chemical composition of the four fermented beverages?

Reply. We added statistical analysis information in the material, methods, and results section (Table 3). We are very grateful for this comment.

Table 3. Characterization of fermented beverages.

Fermented Beverage

Parameters

 Chicha

 Forcha

 Champús

 Masato

pH

3.50 ± 0.08 a

3.76 ± 0.75 ab

3.92 ± 0.15 ab

4.35 ± 0.15 b

Lactic acid (g/L)

16.7 ± 1.7c

8.8 ± 3.0 bc

5.5 ± 2.8 ab

0.3 ± 0.1 a

Ash (%)

2.17 ± 0.82 b

0.46 ± 0.21 a

0.96 ± 0.25 a

0.53 ± 0.01 a

Protein (%)

5.30 ± 0.7 b

1.41 ± 0.3 a

1.12 ± 0.1 a

3.40 ± 0.6 ab

Saccharose (g/L)

4.8 ± 2.9 a

28.1 ± 10.3 b

127.2 ± 18.8 c

117.1 ± 1.8 c

Glucose (g/L)

2.9 ± 2.4 a

23.0 ± 11.5 b

24.0 ± 16.4 b

3.2 ± 0.1 a

Fructose (g/L)

14.1 ± 7.2 b

2.8 ± 1.1 a

21.1 ± 14.0 b

1.0 ± 0.1 a

Soluble solids (%)

8.5 ± 1.5 a

26.3 ± 7.6 c

16.0 ± 4.0 bc

12.5 ± 2.5 ab

Ethanol (%)

9.9 ± 1.2 c

2.9 ± 1.1 bc

0.5 ± 0.2 a

0.2 ± 0.1 a

Results are expressed as means ± standard deviations of three independent samples, each with three replicates. Different letters in the same row indicate significant differences (P < 0.05).

2.18. Table 4: Why did not the authors statistically compare the data of the antioxidant capacity (DPPH and ABTS), total phenolic content (TPC) of the four Colombian fermented beverages?

Reply. We added statistical analysis information in the material, methods, and results section (Table 4). We are very grateful for this comment.

Table 4. Antioxidant capacity (DPPH and ABTS), total phenolic content (TPC) of Colombian fermented beverages.

Beverage

DPPH (µM TE/L)

ABTS (µM TE/L)

TPC (mg GAE/L)

 Chicha

15.24 ± 3.76 b

15.48 ± 1.05 b

546 ± 33 b

 Forcha

3.08 ± 1.28 a

0.55 ± 0.42 a

0.87 ± 0.33 ac

 Champús

1.99 ± 0.45 a

13.13 ± 9.19 b

258 ± 76 bc

 Masato

2.20 ± 0.89 a

0.38 ± 0.20 a

0.43 ± 0.15 a

TE: Trolox equivalent. Results are expressed as means ± standard deviations of three independent samples, each with three replicates. Different letters in the same column indicate significant differences (P < 0.05).

The above-mentioned methodological errors (in the materials and methods section and Results section) should be appropriately addressed to provide a sounder and more profound discussion of the results to give appropriate conclusions supported by a rigorous analysis of the results obtained.

Round 2

Reviewer 1 Report

No more comments.

Reviewer 2 Report

After reviewing both the revised paper and the responses provided by the authors, I consider that the manuscript is now acceptable for publication in Fermentation. The authors' answers are very clear and respectful and they appropriately addressed the comments of the reviewer to improve the manuscript. They have responded to the different questions raised by the reviewer in a very polite and professional way. Congratulations!!